# Impact damage reduction of woven composites subject to pulse current

Yan Li[1], Fusheng Wang [1]✉, Chenguang Huang [1]✉, Jianting Ren[1], Donghong Wang[2], Jie Kong [3], Tao Liu [4] & Laohu Long[5,6]

3D orthogonal woven composites are receiving increasing attention with the ever-growing market of composites. A current challenge for these materials' development is how to improve their damage tolerance in orthogonal and layer-to-layer structures under extreme loads. In this paper, a damage reduction strategy is proposed by combining structural and electromagnetic properties. An integrated experimental platform is designed combining a power system, a drop-testing machine, and data acquisition devices to investigate the effects of pulse current and impact force on woven composites. Experimental results demonstrate that pulse current can effectively reduce delamination damage and residual deformation. A multi-field coupled damage model is developed to analyze the evolutions of temperature, current and damage. Parallel current-carrying carbon fibers that cause yarns to be transversely compressed enhance the mechanical properties. Moreover, the microcrack formation and extrusion deformation in yarns cause the redistribution of local current among carbon fibers, and its interaction with the self-field produces an obvious anti-impact effect. The obtained results reveal the mechanism of damage reduction and provide a potential approach for improving damage tolerance of these composites.

Carbon fiber reinforced composites are extensively used in aerospace, aeronautics, and national defense fields due to its high specific stiffness, strength, and designability[1–4]. However, they are sensitive to low-velocity impact events from dropped tools, debris or birds during its manufacture and service, which can cause delamination failure and result in a significant reduction of load-bearing capacity[5,6]. Numerous studies have therefore been carried out to improve low-velocity impact response of composites from the structural design such as ply-stacking sequence[7,8], ply thickness design[9,10], 3D structure design[11,12]. Compared to 2D laminated composites, 3D orthogonal woven composites (3DOWCs) can improve the delamination behavior due to the presence of through-thickness Z-binder yarns[13–16]. In addition, fiber hybridization is also an effective method of improving impact resistance, as reported in Refs. 17,18. These methods can only be performed in the manufacturing stage as the internal structure and constituent material of the end product are very difficult to be altered again.

By taking advantage of the conductive carbon fibers due to their internal near-graphite structures and the principle of coupling electrical conductors to electromagnetic environment, a certain degree electromagnetic environment can improve the strength and resistance to debris-induced fracture or delamination of the exposed carbon fiber/epoxy laminate[19–22]. The conductivity of carbon fibers can be understood through classical hybrid orbital theory[23,24], as shown in Fig. 1a–d, carbon atoms are arranged in a hexagonal lattice in a plane to

[1]School of Mechanics, Civil Engineering and Architecture, Northwestern Polytechnical University, 710129 Xi'an, PR China. [2]Shanxi Key Laboratory of Electromagnetic Protection Material and Technology, The 33th Institute of China Electronics Technology Group Corporation, 030032 Taiyuan, PR China. [3]Shaanxi Key Laboratory of Macromolecular Science and Technology, School of Chemistry and Chemical Engineering, Northwestern Polytechnical University, 710072 Xi'an, PR China. [4]School of Engineering and Materials Science, Queen Mary University of London, Mile End Road, London E1 4NS, UK. [5]State Key Laboratory of Long-Life High Temperature Materials, 618000 Deyang, PR China. [6]Dongfang Electric Corporation Dongfang Turbine Co.,LTD, 618000 Deyang, PR China. ✉e-mail: fswang@nwpu.edu.cn; huangcg@nwpu.edu.cn

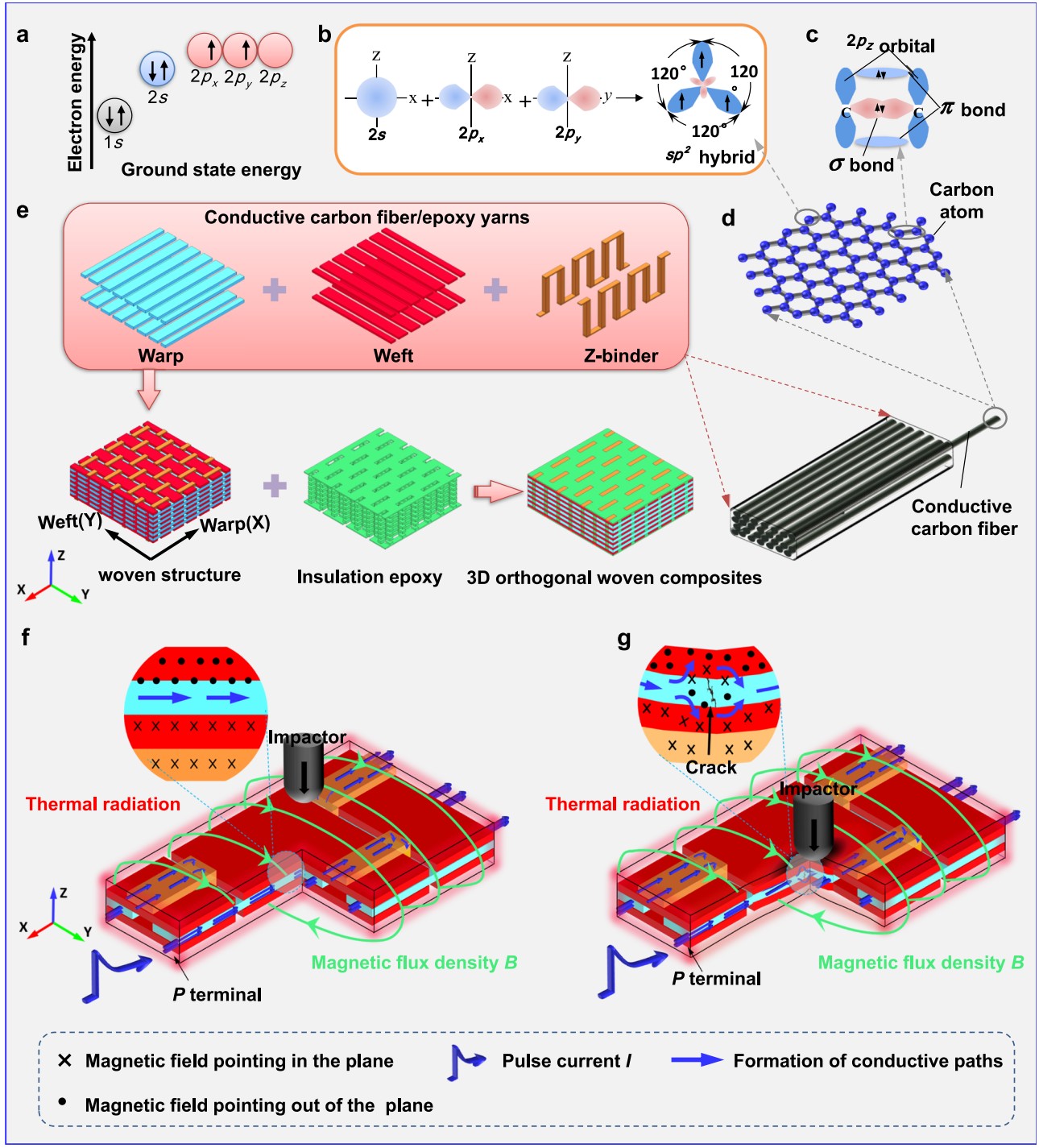

**Fig. 1 | 3D orthogonal woven composite structure and multi-physics field coupling. a** Ground state energy levels of electrons in carbon atoms. **b** $sp^2$ hybrid orbitals. **c** $\sigma$ and $\pi$ bonds built between two carbon atoms by $sp^2$ hybridization. **d** Conductive carbon fiber and its internal near-graphite structure. **e** Structural architecture and its components, including warp, weft, Z-binder, and epoxy resin matrix. **f, g** Dynamic electromagnetic, thermal, and mechanical field responses of 3D orthogonal woven composites subject to pulse current before and after impact.

form a single layer of graphene. Due to a ground state electron configuration of $1s^2 2s^2 2p_x^1 2p_y^1 2p_z^0$ in a carbon atom, three stable in-plane $\sigma$ bonds are made up by $sp^2(2s, 2p_x, 2p_y)$ hybridized orbitals, while free-moving $\pi$ bond located vertically to the lattice plane is made up by $2p_z$ orbitals in graphite structure. The deformation and separation of the hexagonal carbon rings require high energy due to $\sigma$ bonds, providing the strength of the carbon fiber at macro level, while $\pi$ bond makes it a good electrical conductor[25,26]. In addition, the advanced control techniques may couple the structural capabilities conductive composites

with electrical, magnetic, or thermal functions, providing rich possibilities for multifunctional platforms. For example, electromagnetic launch technology has been used in military confrontations in electromagnetic catapults[27], electromagnetic railguns[28], and electromagnetic coilguns[29]. Electric vehicle charging on electrified roads has made use of wireless power transfer technologies[30]. These technologies mainly take advantage of the positive influence of electromagnetic effect on conductive solids and the precise control and diagnosis of structural and electromagnetic systems by information flow. Hence, it

is a perspective to improve the performance of composite materials to make full use of the multifunctional property of materials with advanced control techniques, such as a key physical effect in conductive fibers whereby matter becomes compressed under the effect of an electric field.

Although electromagnetic environments can improve the mechanical properties of conductive laminates, its effect and mechanism on more complex conductive structures such as the 3DOWCs have not been reported. As shown in Fig. 1e, 3DOWCs are comprised of complex woven architecture with conductive carbon fiber/epoxy yarns and insulating epoxy resin filled woven architecture. The conductive warp and weft yarns are interwoven vertically in the plane direction. The conductive Z-binder yarns undulate along warp direction and are applied to bind weft and warp yarns in thickness direction. At this point, the yarns are equivalent to bare wires when pulse current is applied. The currents pass through the interwoven strands of the yarn, forming a complex conductive network as shown in Fig. 1f. The impact force is applied to 3DOWCs on the basis of electrification, which is a typical real-time coupling problem among the electromagnetic, thermal, and mechanical fields, as shown in Fig. 1g. In this case, the Lorentz force due to the current-field interaction between current and its self-field as well as the corresponding electrothermal stress affect the mechanical response. In turn, deformation and damage induced by impact force change the internal structure of the 3DOWCs plate, resulting in the redistributions of electromagnetic and thermal fields.

In this paper, the effects of impact force and pulse current on the 3DOWCs plate and the interaction mechanism of multi-physics field are studied. An integrated experimental platform, which is composed of current supply, drop-testing machine, and data acquisition devices, is designed to realize the synergistic effects of pulse current and impact force on the 3DOWCs plate via wireless telecommunication technology. The dynamic numerical model of synergistic responses for 3D yarn-level orthogonal woven composites with pulse current is developed based on the real-time sequential coupling of electromagnetic, thermal, and mechanical fields. We find that the pulse current introduced into the 3DOWCs plate can effectively reduce impact damage. Our experimental observations and simulation results further reveal the damage reduction mechanism for the 3DOWCs under pulse current and impact load.

## Results

### Design of the integrated experimental platform
The experiments are carried out on a self-developed pulse current-impact and data collection integrated experimental platform at ambient temperature. As shown in Fig. 2a, the platform includes a DIT152 drop-testing machine, a function signal generator (Puyuan DG4062), a current supply (Agilent 6692A), a Tektronix MDO3054 oscilloscope, a current probe (TCP0150), a voltage probe (DP1650A), a set of data acquisition devices and a self-developed computer-controlled trigger system. This system allows the drop falling time, current action time and data collection time to be controlled arbitrarily. It is noted that Agilent 6692A 6600-watt power supply provides a steady current, while a time-varying pulse current is required in the experiment to reduce the adverse effect of the current-induced Joule heating on the plate. To this end, a function signal generator (Puyuan DG4062) is connected to the current supply to generate arbitrary time-varying voltage waveforms. Once the two devices are connected, current supply trigger and the generated current waveform are controlled by the function generator. The input voltage $U_{in}$ on the function generator and the output current $I_{out}$ on the power supply can be calculated via the relationship $U_{in}/5 = I_{out}/110$.

In addition, a fixture must be electrically insulated and meet the impact requirements in accordance with ASTM D5379 to characterize composite material correctly during measurement. As shown in the

upper left corner of Fig. 2a, a fixture is customized and constructed according to the standard. Non-conductive wood is utilized to construct the main body. Copper bars are used to construct the positive and negative electrodes, ensuring that the current is conducted along the composite in the experiment. The 3DOWCs plates with dimensions of 148 mm × 148 mm × 4.5 mm are made of F-46 epoxy resin and T700 carbon fiber with a fiber volume fraction of 82.43%, as shown in Fig. 2b. Composite prepreg contains 6 layers of warp tows, 7 layers of weft tows and the binding yarns, each of which is composed of 12 K carbon fibers. A 3DOWCs plate is a part of the circuit and the contact resistances between the plate and copper bus bars are negligibly small by evenly coating a thin layer of conductive silver paint (SPI#05002-AB) at their interfaces.

Before starting the collaborative experiment of impact force and pulse current on the 3DOWCs plate, all the devices need to be connected and adjusted to the required parameters, as shown in Fig. 2c. The impact module is relatively integrated in the drop-testing machine and thus its connection is ignored. In order to realize signal conversion, the function generator's positive output wire and negative output wire are connected to the VP connector and IP connector located on the back of the power supply, respectively. The 3DOWCs plate is centered in the customized fixture to ensure that the impact point is exactly centered on its upper surface. Two lead wires of the copper electrodes from the power supply are in contact with the two sides of the 3DOWCs plate. As shown in Fig. 2c, the devices are connected to red lines to form a series circuit. The composite resistance is obtained by Ohm's law. The voltage probe is fixed on two copper electrodes to measure the voltage across the 3DOWCs plate. The current probe is held on the cable, which connects the power supply and copper electrode to measure the current passing through the 3DOWCs plate. The voltage and current data are recorded by the oscilloscope. Taking into account the anisotropy of the 3DOWCs plate, two heat-flow sensors are attached to the plate along the warp and weft directions to detect the heat flow and temperature responses. As shown in Fig. 2a, eight thermal resistors are also attached around impact point of the plate to detect temperature changes around the impact point. The heat flow data are collected by a heat flow meter and the thermal resistance data are collected by a 20-channel data acquisition instrument. Once all the devices are connected, the required pulse waveform and amplitude, as shown in Fig. 2d, are set through the mechanical buttons on the function generator. Finally, the trigger time of impact module, current module, and data acquisition module in the self-developed control program are modified according to the experimental requirements. Note that the duration of the impact force is more than three orders of magnitude shorter than that of the pulse current. Thus, in order to make sure that the impactor just contacts with the specimen when the pulse current reaches its peak value, the impact module is controlled to delay triggering for a time of $t_* - t_f$. $t_*$ is the time when the current reaches its peak. $t_f = \sqrt{2h/g}$ is the free-fall time of the impactor, where $h$ denotes the height of the gravity center of the impactor from the upper surface of the plate, and $g$ denotes the gravitational acceleration. In this step, the coordination of experiment and acquisition devices using wireless communication technology is required, which is key to the success of the integrated experimental platform. After confirmation, the run button in the program is triggered to make each device start working.

### Pulse current-induced reduction effect on impact damage
Time histories of force, deformation, and impact energy corresponding to impact energy of 50 J and different pulse current peaks (i.e., $I_{am} = 0$ A, 30 A, 70 A, and 110 A) are plotted in Fig. 3a, b, and c, respectively. In the initial loading stage (Stage I), the impactor begins to contact with the plate and then leads to local deformation near the impact point. Meanwhile, the kinetic energy of the impactor is gradually converted into the elastic energy of the plate. As small elastic

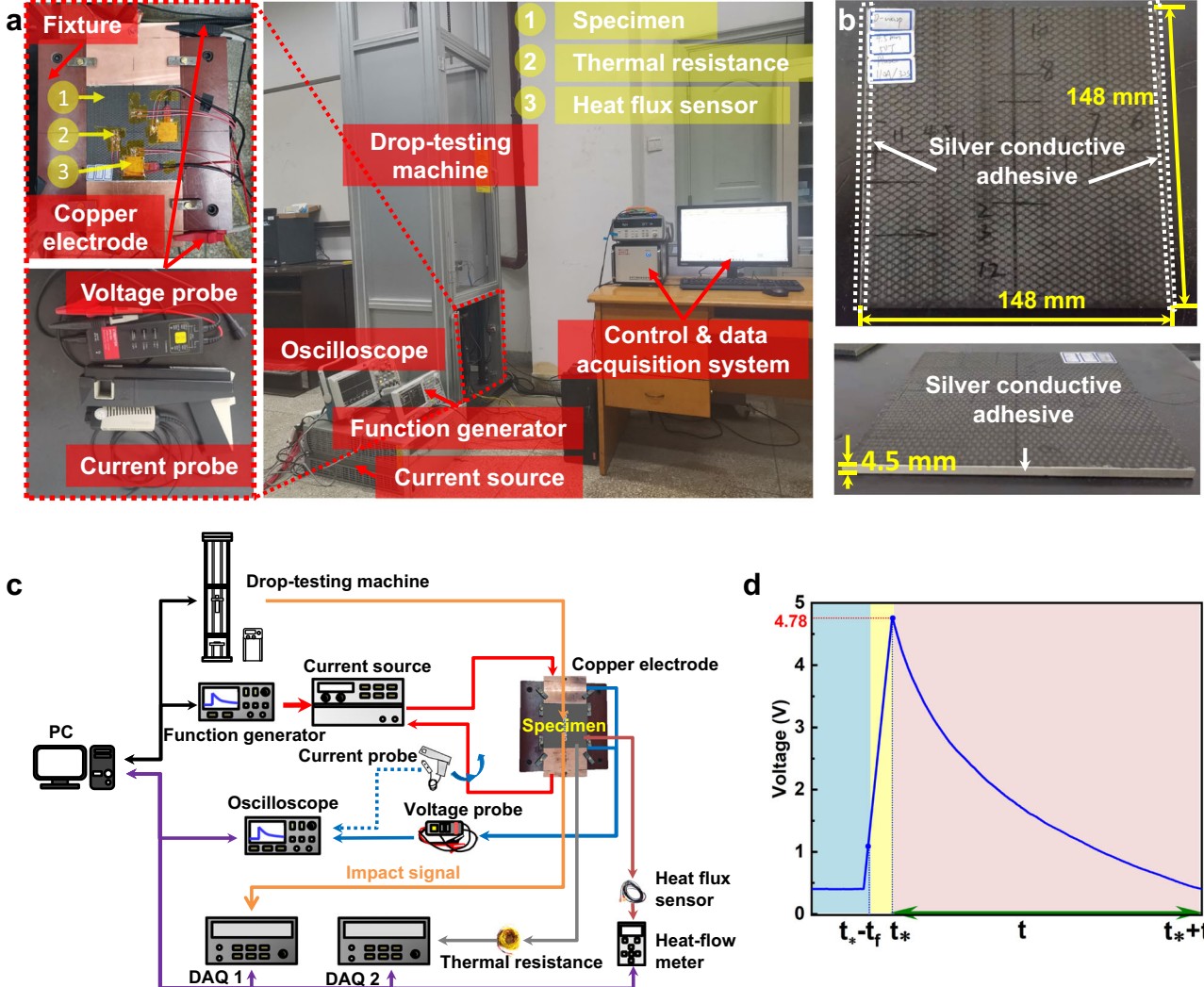

**Fig. 2 | The designed experimental platform and specimen. a** The integrated experimental platform of drop impact, current and test collection. **b** Specimen size and conductive adhesive distribution. **c** Schematic of device connections. **d** Double exponential pulse voltage waveform.

deformation hardly affects the current distribution, the force, deformation or energy curves corresponding to different $I_{am}$ almost overlap in this stage. Then, the impact force increases rapidly and some oscillations occur due to multiple factors, such as large difference in mass or stiffness between the impactor and the plate. As inelastic energy accumulates, the plate gradually enters the maximum loading stage (Stage II). Large deformation is observed near the impact point, and the plate is strongly deformed locally, resulting in densification of the current channels, which in turn resists the impact through the Lorentz force generated by the current interacting with its self-field. In this stage, the force curves are separated, and so do the deformation and energy curves. When the plate reaches the maximum deformation, the kinetic energy of the impactor is completely converted into the elastic and inelastic energies of the plate. At the same time, the plate is subjected to the maximum impact force. It is found that as $I_{am}$ increases, the maximum impact force increases gradually. Besides, the maximum energy in the plate appears at $t$ = 3.312 ms, 3.224 ms, 3.196 ms, and 3.188 ms for the cases of $I_{am}$ = 0 A, 30 A, 70 A, and 110 A, respectively, and the corresponding maximum deformations are also observed at the same instants. That is, the plate will arrive early at the most dangerous stage of damage with increasing $I_{am}$. In the rebounding stage (Stage III), the impactor is rebounded by the plate until the two are completely separated. As the impact force decreases, the plate gradually recovers its deformation due to the release of

elastic energy, while the residual deformation is remained due to the dissipation of inelastic energy. Moreover, the inelastic energy or residual deformation decreases with the increase of $I_{am}$, which can be observed more intuitively from Fig. 3d. The inelastic energy and residual deformation of $I_{am}$ = 110 A are respectively reduced by 35.81% and 47.64% compared with those of $I_{am}$ = 0 A, implying that the Lorentz force generated by the pulse current in the warp direction can effectively resist the impact force on the plate.

An optical microscope and ultrasonic C-scan device are adopted to characterize the effect of pulse currents on the impact damage reduction of the plates. The damage morphology for the cross section of the plate under impact energy of 50 J is shown in Fig. 3e. The resin matrix around the impact point is debonding from the top and bottom surfaces of the plate. This is because the resin matrix is in an unconstrained state and its mechanical performance is inferior to that of the yarns. Delamination among various types of yarns is found on the bottom surface of the plate due to debonding or even shedding of the resin matrix. In addition, resin cracks among yarns are found under the impactor, which are blocked by the interwoven yarns to effectively suppress the propagation of cracks. The delamination depth and area after impact are obtained to quantify the damage reduction efficiency of pulse current on the plate, which is based on the conventional B-scan and ply-by-ply C-scan analysis in the ultrasonic C-scan device. The gross damage images with a resolution of 1500 × 1500 pixels are

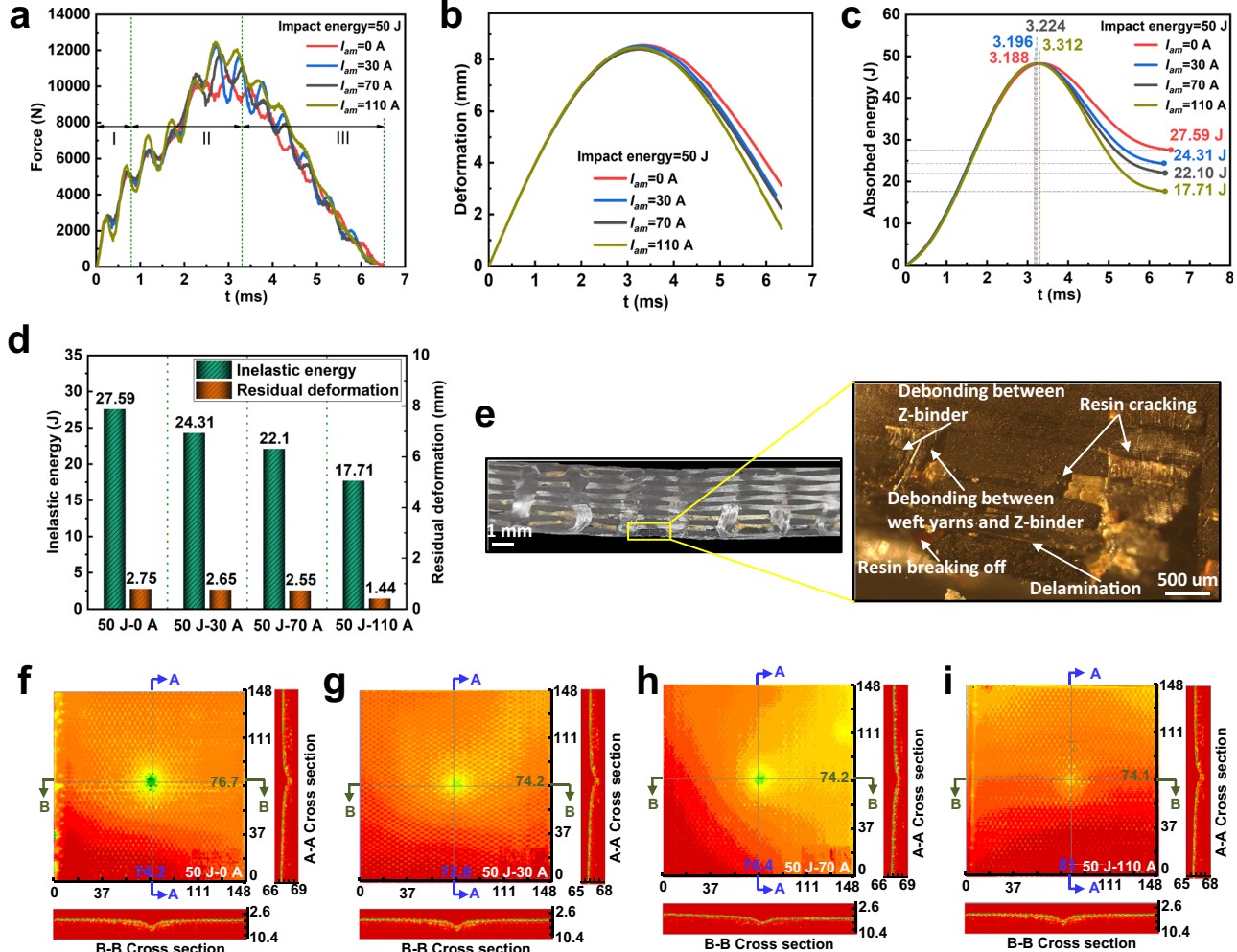

**Fig. 3 | Experimental and detection results for 3D orthogonal woven composite subject to impact energy of 50 J and different pulse currents. a–c** Impact response curves of force, deformation, and absorbed energy. **d** Inelastic energy and residual deformation. **e** Microscope images of cross sections with pulse current peak 0 A. **f–i** Ultrasonic C-scan images with different pulse current peaks 0 A, 30 A, 70 A, and 110 A.

shown in Fig. 3f–i for the plates under impact energy of 50 J with $I_{am}$ = 0 A, 30 A, 70 A, and 110 A. The depth of damage is illustrated by color, ranging from red (shallow) to green (deep). At the cross sections of A-A and B-B, the green circular area with damage depth greater than 1 mm accounts for 0.057% of the full-scale 3DOWCs plate for the case of $I_{am}$ = 0 A. As $I_{am}$ increases, the green area gradually shrinks, indicating a significant reduction of the impact damage. Since few experimental tools are available to directly observe the multi-field coupling evolution inside composite structures, the reduction mechanism of pulse current on impact damage will be further discussed in detail by means of numerical models.

**Evolution of multi-physics field**

To further investigate the mechanism of damage reduction, the evolutions of multi-physical fields and impact damage are presented in Figs. 4 and 5 for the plate under impact energy of 50 J and $I_{am}$ = 110 A, respectively. Before impact, the yarns in the plate are spatially interlaced horizontally and vertically to form an overlapping 3D network with interface resistance. When the two ends of the plate are connected to a circuit, the electric potential decreases uniformly along the warp direction, and the current density through the YZ cross section is evenly distributed at the macro level. The interaction of the current with its self-field causes the plate to be subjected to a compressive electromagnet force. In the initial loading stage, the kinetic energy of

the impactor begins to convert into the elastic energy of the plate, and the deformation of the plate increases with the impact force. At the moment, the potential difference around the impact point increases due to the impact deformation (see Fig. 4). The yarns around the impact point are squeezed, resulting in a higher current density than elsewhere. Meanwhile, the enhanced magnetic flux density around the impact point can be observed in the XY plane, resulting in stronger Lorentz force in this region. The temperature change is insignificant during the whole process because the highest increase is only 0.16 °C due to the tiny peak and short duration of the pulse current. This feature differs greatly from that of direct or alternating current, where the electrothermal effect caused by the transport current can lead to severe damage inside laminates, as illustrated in ref. 22.

In the maximum loading stage, energy of the impactor is continuously converted into elastic and inelastic energy of the plate as shown in Fig. 5. Such behavior leads to resin matrix cracking, debonding, and yarn breakage around the impact point because the stress induced by impact deformation of the plate reaches the damage tolerances of the resin and yarns. At this point, large deformation and damage lead to changes in the current channels. However, the overall current is still concentrated in the deformed warps, although the interface resistances become smaller due to the extrusion among the components. The Lorentz force generated by the redistributed current and its self-field continuously resists the impact force. The physical

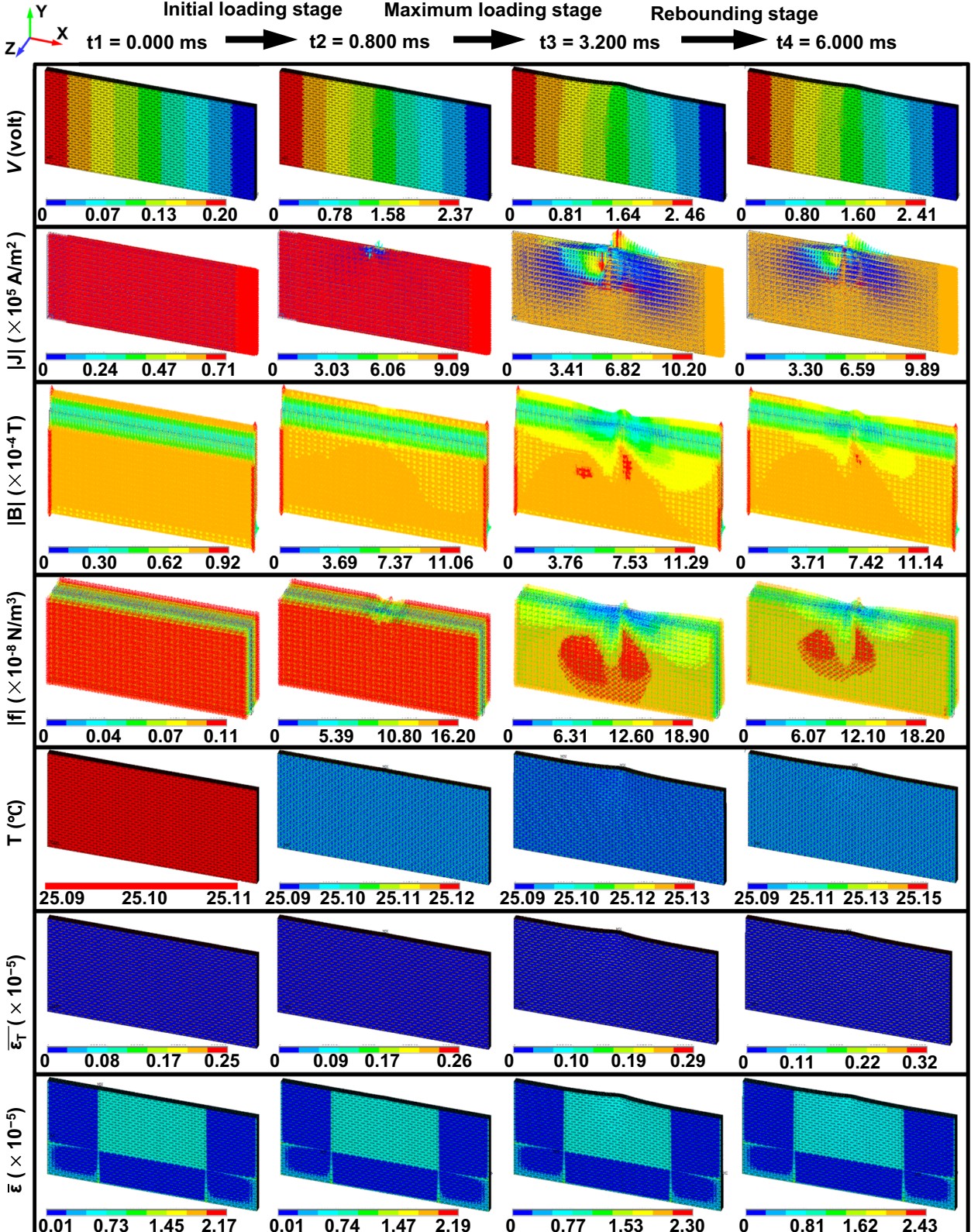

**Fig. 4 | Responses of 3D orthogonal woven composite under impact energy of 50 J and pulse current peak 110 A at the selected time points, marked in Supplementary Fig. 2l.** The figure shows the distributions of electric potential, current density, magnetic flux density, electromagnetic force, temperature, effective thermal strain, and total effective strain from top to bottom.

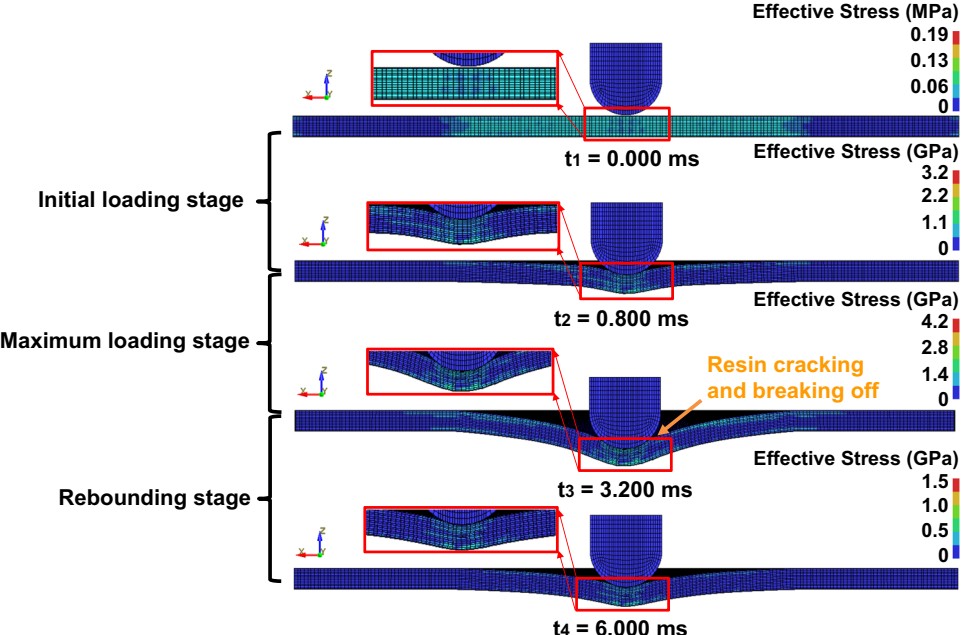

**Fig. 5 | Impact response of 3D orthogonal woven composite under impact energy of 50 J and pulse current peak 110 A at the selected time points, marked in Supplementary Fig. 2l.** As the energy of the impactor is converted to the elastic and inelastic energy of the plate, yarns are squeezed and even damage, indicating that large deformation and damage lead to changes in current channels.

mechanism the damage reduction applied pulse current is Ampere's Law for a single yarn and 3D orthogonal woven composites. For a single yarn, carbon fibers in the yarn are equivalent to conductive wires when a current is applied (see Fig. 6a). These parallel carbon fibers produce electromagnetic forces that are pointing toward the center of cross-sectional area, in accordance with Ampere's Law. Although a single carbon fiber's electromagnetic force is very small, the collective effect of 12,000 carbon fibers can produce a force large enough to cause macroscopic transverse contraction deformation. Figure 6b shows the normalized displacement of a single yarn under the same external load with currents of 0 A, 0.14 A, 0.32 A, and 0.51 A, respectively. An obvious decrease in the generated normalized displacement is observed with pulse currents, indicating that current can improve the mechanical properties of yarn, as has been proved by previous work[31,32]. When the impact point of the plate is moved downward by impact force for a 3D orthogonal woven fabric, the local upper part along the thickness gradually moves below the mid-plane and the direction of the electromagnet force changes from downward to upward, which produces a macroscopic resistance just opposite to the impact force, as shown in the side view of Fig. 6d. As shown in the top view of Fig. 6e, the upper and lower parts of the plate along the width are subjected to vertical downward and upward electromagnetic forces, respectively. Under the impact force, the upper part of the warp yarns around the impact point gradually bend upwards, while the lower part of the warp yarns bend downwards, which results in the increase of the density of local warp yarns. The warp yarns are then subjected to a large electromagnetic force directed towards the impactor, thereby resisting the impact force. It should be noted that when the warp yarns are partially cracked or even broken due to excessive impact (see Fig. 6c), the current will form a new circuit in the intricate yarns to continuously resist the impact force. At this time, the maximum effective thermal strain value increases, but still contributes less than the compressive electromagnetic strain generated by Lorentz force.

In the rebounding stage, the plate cannot recover to its original state because the damage and plastic deformation has dissipated part of the kinetic energy of the impactor. At this time, the extrusion effect among yarns is weakened, reducing the magnitude of pulse current density. However, the direction of the current density in this stage remains consistent with that in the maximum loading stage. Meanwhile, the magnetic flux intensity around the impact point decreases compared with that in the maximum loading stage. As a result, the magnitude of the total Lorentz force decreases while its direction coincides with the rebound direction of the impactor. In this sense, the total Lorentz force at this stage can accelerate the rebound of the plate. Moreover, the contribution of the Lorentz force is much larger than that of the effective thermal stress, further indicating that the pulse current can reduce the impact damage of composite plates.

## Discussion

This work provides a strategy to reduce impact damage in 3D orthogonal woven composites by combining the structural property of woven fabrics and electromagnetic property of carbon fibers. An integrated experimental platform is designed to study the synergistic effects of pulse current and impact force by means of wireless telecommunication technology. The experimental results show that the introduction of pulse current to 3D orthogonal woven composites can significantly reduce the depth and area of impact damage. To be specific, an increase in pulse current peak from 0 to 110 A results in a decrease in inelastic energy from 27.59 to 17.71 J, as well as a reduction in residual deformation from 2.75 to 1.44 mm. The mechanism of damage reduction is revealed by using a multi-field coupled model on the basis of the theories of damage mechanics extended to account for classical electromagnetism and heat transfer. For a single yarn, the collective effect of Lorentz forces generated by parallel current-carrying carbon fibers leads to macroscopic transverse contraction deformation, implying that current can improve the mechanical properties. For the whole 3D orthogonal structure weaved by yarns under impact loading, the local extrusion among yarns changes the direction of pulse current flowing in carbon fibers, and its interaction with the self-field just provides a compressive electromagnetic force that resists the impact force. The current will form a new circuit in the intricate yarns to continuously resist the impact even if the impact

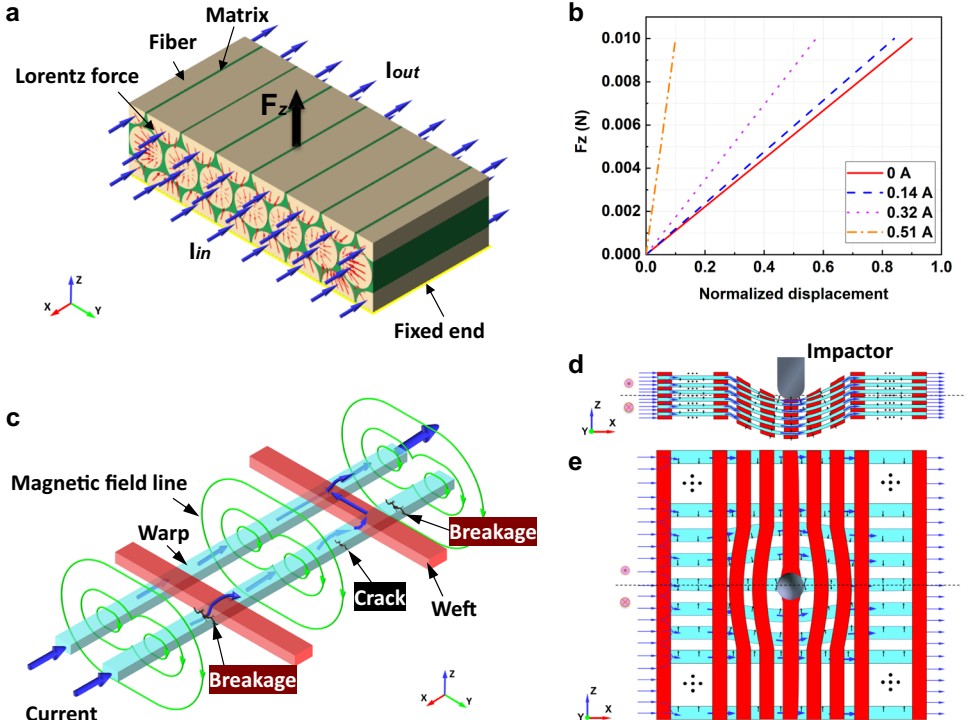

**Fig. 6 | The impact damage reduction mechanism in 3D orthogonal woven composites subject to pulse current. a** Schematic of electric current flowing through the single loaded yarn, in which the yarn contains carbon fiber and matrix. $F_z$ is the external load applied to the upper surface of the yarn perpendicular to Z axis. **b** The change of tensile force with respect to min-max normalized displacement for single yarn under different pulse current peaks 0 A, 30 A, 70 A, and 110 A. **c** Schematic of electric current and magnetic field in the energized interlaced yarns. **d, e** The distributions of Lorentz force within the XZ and XY planes during impact process. Blue and black arrows represent the directions of pulse current and Lorentz force, respectively.

damage causes a partial circuit break. In addition, the pulse current can keep the energy loss of composites at a low level, which effectively avoids damage caused by thermal expansion. Based on the results, we provide some general guidelines on how to take full advantage of the electromagnetic property of carbon fibers within composites to achieve active impact damage reduction.

## Methods
### Multiscale modeling of 3DOWCs
The Mechanical APDL Product Launcher module of ANSYS 19.0 software package is used for thermal-mechanical behavior study of the yarn. As schematically shown in Supplementary Fig. 1a, 3DOWCs possess the naturally multiscale characteristics, such as thousands of fibers embedded in epoxy resin at the microscale, various yarns and inter-yarn epoxy resin at the mesoscale, and lots of mesoscale RVE at the macroscale. The synergistic responses of 3DOWCs plate are studied sequentially by a three-scale modeling strategy including microscale, mesoscale, and macroscale models.

The classical hexagonal microscale RVE model with temperature-dependent damage of each component is established to obtain the mechanical and thermal input parameters for the yarn based on the homogenization theory[33–35]. The brittle behavior and temperature-independent below 200 °C[36] for transversely isotropic carbon fibers are characterized by the maximum stress criterion. The elastic constants and strengths of carbon fibers are provided by the manufacturer and listed in Supplementary Table 1. A temperature-dependent elastic-plastic damage model is modeled to characterize resin matrix behavior, which follows the theory of bilinear isotropic hardening and von Mises yield criterion. The experimental data of resin matrix are shown in Supplementary Fig. 1b–d. In addition, a bilinear cohesive model governed with a traction-separation law is introduced to characterize the damage behavior of fiber/matrix interface (see Supplementary

Fig. 2a). Since the lack of interface properties for carbon fiber/matrix (T700/F-46), a set of experimentally calibrated interface parameters of similar material system (T700/TF1408) are used here to describe the change of interface properties with temperature[37]. The kink force $F_d$ and maximum pull-out force $F_{max}$ measured by microbond test at different temperatures are listed in Supplementary Table 2. The interface stiffness, strength, and fracture toughness varying with temperatures are listed in Supplementary Table 3. The results of the yarn using finite element method match well with those of the mixture rule method and Chamis model[38] at 25 °C, indicating that the microscale model is suitable for the mechanical response analysis at different temperatures (see Supplementary Table 4). Based on the proposed microscale classical hexagon model with interfacial damage of fiber/matrix, the stress-strain curves and failure modes of microscale RVE at 25, 60, 80, and 120 °C are shown in Supplementary Fig. 2c–h. In the longitudinal direction, as temperature rises, the interface properties of strength and fracture toughness decreases so that the degree of interface slip increases, which is shown in Supplementary Fig. 2b. However, the longitudinal tensile and compressive stiffness and strength change little, because the failure is mainly controlled by temperature-independent carbon fibers. For the transverse tension and compression, the properties are mainly determined by resin matrix and interface which alter with temperature. As shown in Supplementary Fig. 2e, f, stress-strain curves exhibit a trend of first increasing and then decreasing, which corresponds to the interface damage evolution represented by the traction-separation curve of interface. It should be noted that interface failure leads to the final failure of RVE in transverse compression as the temperature rises. For in-plane shear, stress-strain curves present typical elastic-plastic properties and the failure modes become the damage of resin matrix and interface. For out-plane shear, the final failure is governed by the shear failure of resin matrix. The out-plane shear properties decline

with the resin softening at higher temperature, and the failure modes at different temperatures are almost similar.

The current conduction in yarns exhibits strong anisotropy and is temperature-independent below 343 °C, which have been validated by experiments[39–43]. It is worth noting that fibers in yarns are in contact with each other and form a contact network in transverse direction due to slight inclinations or ripples. In this sense, the hexagon model applied to calculate mechanical and thermal parameters is unsuitable for calculating transverse conductivity. Therefore, the mixture rule and Kirchhoff's current law in refs. [44,45] are used to calculate the longitudinal and transverse conductivities, respectively. The electrical parameters of carbon fiber are given in Supplementary Table 5 and the obtained conductive parameters of yarn are listed in Supplementary Table 6 with a fiber volume fraction of 82.43%. Carbon fiber and epoxy resin are non-magnetic materials so that the permeability of yarns is $1\mu_O$.

In the mesoscale, a single RVE is established based on the geometric structure of the 3DOWCs plate and the microscopic photographs of cross sections as shown in Supplementary Fig. 4e, f. The mesoscale single RVE contains regularly arranged yarns and resin matrix filled among the yarns. The Chang-Chang criterion[46] and elastoplastic model are developed to investigate the damages of yarns and resin matrix, respectively. The cross sections of all yarns are assumed to be rectangular to guarantee the numerical convergence. Both the weft and warp yarns are straight and perpendicular to each other with interlacement. Furthermore, Z-binder yarns undulate along the warp direction with a useful feature to bind weft and warp yarns in the thickness direction. Then, a macroscale model with the size of 148 mm × 74 mm × 4.5 mm is established by duplicating the mesoscale single RVE.

## Construction of multi-physics field interaction model

Multi-field coupled simulations are carried out using Mechanical APDL Product Launcher module of ANSYS 19.0 software package. In order to form a conductive path across the model, pulse current is imposed to one side of the model and the potential on the opposite side is set to 0 V. Heat radiation with a coefficient of 0.9 is applied to the top and bottom surfaces exposed to the air. Meanwhile, the parallel magnetic flux boundary is applied to the top and bottom surfaces and the opposite side of symmetry plane for the 3DOWCs plate. The vertical magnetic flux boundary is applied to the symmetry plane of the 3DOWCs plate. The ambient temperature is set to 25 °C. In addition, the simply supported boundary and fixed boundary of the plate are consistent with those in the experiment. The hemispherical impactor with a diameter of 16 mm and a mass of 2.75 kg is modeled as a rigid body. Moreover, all degrees of freedom of the impactor except axial direction are fixed to avoid rotating or shifting. The initial velocity is defined to the impactor reference point according to the initial impact energy requirement. Furthermore, the cyclic sequential coupling method is adopted to solve the interaction of electromagnetic, thermal, and mechanical fields on the 3DOWCs plate. Electromagnetic and thermal processes are conducted using implicit solving methods, while the low-velocity impact process is carried out using explicit solving strategies. In order to verify the multi-physics field coupling model, the numerical force curves for impact energy of 50 J with $I_{am}$ = 0 A, 30 A, 70 A, and 110 A are compared with the experimental force curves. These two sets of curves exhibit nearly the same trend (Supplementary Fig. 2i–l). In addition, the maximum impact forces $F_{max}$ obtained by numerical and experimental methods are listed in Supplementary Table 7, in which the two types of data are in good agreement with a mean error of 15.09%.

## Data availability

All data generated in this study are provided in the Supplementary Information/Source data file or from the corresponding author upon request. Source data are provided with this paper.

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

## Acknowledgements

This work is supported by the National Natural Science Foundation of China (Grant numbers 51875463 and 52175147) to F.W., Aviation Science Foundation of China (Grant number 20200044053002) to F.W., the special fund for Science and Technology Innovation Teams of Shanxi Province (Grant number 202204051001001) to D.W and the fund of State Key Laboratory of Long-life High Temperature Materials (Grant number DTCC28EE200788) to J.R.

## Author contributions

Y.L. designed the integrated experimental platform, performed the experiments, and wrote the manuscript. F.W. and C.H. conceived the original idea, developed the numerical model, contributed to the data analysis, wrote the manuscript and scientific discussions. J.R. and D.W. performed the main experimental observations and characterizations. J.K. and T.L. revised the manuscript. L.L. supervised experimental and computational work. All authors discussed the results and commented on the manuscript.

## Competing interests

The authors declare no competing interests.
