## [Peer Review File · Nature Communications]

Impact damage reduction of woven composites subject to pulse currentReviewers' comments:

Reviewer #1 (Remarks to the Author):

The paper reports on an appealing way to reduce (not suppress!) the damage development in a fibre reinforced composite laminate.

The term "suppression" is too optimistic, since damages are induced in the laminate even during high pulsed magnetic field is applied.

The work would be very interesting and fascinating, due to data interpretation and the informative pictures, since laminated composites are very sensitive to high strain rate generated damages. Nevertheless, there is a huge issue, according to this reviewer. The authors focus their discussion on the magnetic field generated counteracting force but, as evident from simulations, the magnitude of mechanical force generated from the magnetic field is way weaker than the stress induced by the indenter during the impact. Hence, the effect of a pulsed force could be not very significant and cannot be the very reason why the impact response is improved at high pulsed current.

A more likely hypothesis could be related to the fact that the applied magnetic field can change the interaction between fibres and matrix, but no comments are made on this hypothesis.

One aspect that the authors did not consider at all is the change in the interface strength between fibers and matrix with the application of an intense pulsed current. The application of a potential between the ends of carbon fibers yarns induce a current flow. The temperature of the fibres changes (as shown in the simulations) with the increase in the current intensity (in particular under the impacted zone) and this can result in localized changes in the polymer/fibre interactions. In particular, the change in temperature affects the fibre/matrix interface strength, as demonstrated by Sorrentino et al in 2017 (<http://dx.doi.org/10.1016/j.compositesb.2017.01.010>) for laminated composites based on woven carbon fibres. In particular, an increase in temperature results in a significant increase in the impact strength, and consequently in the reduction of the damage at constant energy. This phenomenon could verify during the application of an intense current to the laminates and it should be investigated to clarify if it's the fibre/matrix interface strength change or the pulsed magnetic field that is responsible for the reduced (not suppressed) damage in laminates.

For this reason, the paper in the current form, albeit almost well written and structured, cannot be recommended for publication in Nature Communications.

Reviewer #2 (Remarks to the Author):

In this work, a pulsed current was applied to woven composite structures to suppress damage from impact.

1- Please proof read manuscript for typographical errors and proper grammar

2- What is a delta bond and what is the variable π in the introduction. These terms must be better defined

3- In this work, impact testing with a pulsed current experimental setup was performed. In addition a multi-physics FEA simulation was conducted to compare results. This method was able to reduce inelastic energy and deformation to test samples

4- In the experimental program, it is not clear how many samples were tested. Sample size, error bars and other statistical data is required to support experimental findings from the impact tests.

5- The approach outline in the manuscript is interesting but it isn't clear how this method would apply in a real-world setting. In the introduction the authors state that damage to carbon fiber composites is commonly caused by events like tool strikes or bird impacts. From my understanding of the method the timing of the electrical pulse has to coincide with the impact damage event. I'm curious how this method will apply when in the real world timing of damage will be unpredictable and therefore may not occur at the same time of a given electrical pulse. Therefore, I am not sure of the practicality of this method.

Response to the reviewer and editor comments

We have received your comments of our manuscript (NCOMMS-22-27366A-Z). We thank the reviewers and editor for giving us the opportunity to revise our manuscript. According to the reviewers and editor's comments, we have tried our best to revise, explain and reply point by point. All revised places are highlighted by red color in our revised manuscript. We hope that this revised manuscript has addressed all the comments and suggestions.

Reviewers' comments:

Reviewer #1: The paper reports on an appealing way to reduce (not suppress!) the damage development in a fibre reinforced composite laminate.

The term "suppression" is too optimistic, since damages are induced in the laminate even during high pulsed magnetic field is applied.

The work would be very interesting and fascinating, due to data interpretation and the informative pictures, since laminated composites are very sensitive to high strain rate generated damages.

Nevertheless, there is a huge issue, according to this reviewer. The authors focus their discussion on the magnetic field generated counteracting force but, as evident from simulations, the magnitude of mechanical force generated from the magnetic field is way weaker than the stress induced by the indenter during the impact. Hence, the effect of a pulsed force could be not very significant and cannot be the very reason why the impact response is improved at high pulsed current.

A more likely hypothesis could be related to the fact that the applied magnetic field can change the interaction between fibres and matrix, but no comments are made on this hypothesis.

One aspect that the authors did not consider at all is the change in the interface strength between fibers and matrix with the application of an intense pulsed current. The application of a potential between the ends of carbon fibers yarns induce a current flow. The temperature of the fibres changes (as shown in the simulations) with

the increase in the current intensity (in particular under the impacted zone) and this can result in localized changes in the polymer/fibre interactions. In particular, the change in temperature affects the fibre/matrix interface strength, as demonstrated by Sorrentino et al in 2017 (<http://dx.doi.org/10.1016/j.compositesb.2017.01.010>) for laminated composites based on woven carbon fibres. In particular, an increase in temperature results in a significant increase in the impact strength, and consequently in the reduction of the damage at constant energy. This phenomenon could verify during the application of an intense current to the laminates and it should be investigated to clarify if it's the fibre/matrix interface strength change or the pulsed magnetic field that is responsible for the reduced (not suppressed) damage in laminates.

Answer: Thank you for these comments. We agree that the term “suppression” is not appropriate for reporting positive effects of electromagnetic field on composites. We have updated all the term "suppression" to "reduction" in the revised manuscript.

In our experiment, it is observed that with increasing peak of pulse current, the maximum deformation and inelastic energy of 3D orthogonal woven composite decrease, and its bearing capacity increases.

This finding suggests that pulse current has a weakening effect on impact damage. However, as you point out, Lorentz force is weaker than impact force. In order to further reveal the weakening mechanism of pulse current on impact damage, we consider the effects of temperature, fiber/matrix interface and electromagnetic force on a single yarn and 3D orthogonal woven composite. The results show that the maximum temperature rise of the plate is 0.13 °C in our study. Combined with our characterization of the yarn properties considering fiber/matrix interface under the influence of temperature, the matrix/fiber interface can be further excluded to improve the performance of composite plate applied pulse current. We find that the physical mechanism of the damage reduction applied pulse current can be explained from two aspects. For a single yarn, the collective effect of Lorentz forces generated by parallel

current-carrying carbon fibers leads to macroscopic transverse contraction deformation, which implies that current can improve the mechanical properties. For the whole 3D orthogonal structure weaved by yarns under impact loading, the local extrusion among yarns changes the direction of pulse current flowing in carbon fibers, and its interaction with the self-field just provides a compressive electromagnetic force that resists the impact force. The current will form a new circuit in the intricate yarns to continuously resist the impact even if the impact damage causes a partial circuit break. The corresponding explanation has been made in manuscript.

We fully agree with you that the effect of the temperature generated by pulse current on the fiber/matrix interface strength should be considered. We carefully read the literature you provided and found that the resin was thermoplastic in the publication. However, the resin used in our work is thermosetting and its mechanical properties decrease with increasing temperature, which has been confirmed by experiments [1]. In order to study the effect of temperature-dependent fiber/matrix interface properties on mechanical parameters of yarn, we have established a microscale damage representative volume element (RVE) using multiscale method. Further, we have redone all of our analyses using the yarn parameters taking into account fiber/matrix interface effect. The corresponding revision has been made in manuscript. See below for additional details.

As shown in Fig. 4a, RVE of the impregnated yarn consists of carbon fiber, surrounding matrix and fiber/matrix interface. The carbon fiber is transversely isotropic with brittle behavior and temperature-independent below 200 °C [2]. The elastic constants ($E_{f1}, E_{f2}, G_{f12}, G_{f23}, \mu_{f12}$) and strengths (X_{ft}, X_{fc}) of T700 carbon fiber are provided by the manufacturer and listed in Table 1. The maximum stress criterion is used to characterize the brittle fracture of carbon fiber in its longitudinal direction

$$d_f = \begin{cases} 1, & |\sigma_{11}| > X_{ft} \text{ or } X_{fc} \\ 0, & |\sigma_{11}| \leq X_{ft} \text{ or } X_{fc} \end{cases} \quad (1)$$

where $d_f = 1$ denotes that carbon fiber is damaged and $d_f = 0$ indicates undamaged.

A temperature-dependent elastic-plastic damage model is modeled to characterize resin matrix behavior, which obeys the theory of bilinear isotropic hardening and von Mises yield criterion. The temperature-dependent coupled constitutive relation of resin matrix is given by:

$$\boldsymbol{\varepsilon}_m = \mathbf{C}_m^{-1}(T) \boldsymbol{\sigma}_m + \boldsymbol{\alpha}_m(T) \Delta T \quad (2)$$

where $\boldsymbol{\varepsilon}_m$ and $\boldsymbol{\sigma}_m$ are resin matrix strain and stress tensors, respectively; \mathbf{C}_m is the stiffness matrix with two independent elastic constants (E_m, ν_m) , which are tested at different temperatures and shown in Fig. 4b; $\boldsymbol{\alpha}_m$ is vector of coefficient of thermal expansion, which is also tested at different temperatures and shown in Fig. 4d; ΔT is the difference between the reference temperature T_0 and current temperature T . Once the plasticity occurs, a bilinear isotropic hardening plastic model is used to characterize the plastic behavior. The temperature-related yield function $f(\boldsymbol{\sigma}(T))$ is given by

$$f(\boldsymbol{\sigma}(T)) = \sigma_0(T) + E_p(T) \boldsymbol{\varepsilon}_{eff}^p \quad (3)$$

where $\sigma_0(T)$ and E_p are the initial yield stress and tangent modulus, respectively. $\boldsymbol{\varepsilon}_{eff}^p$ is the effective plastic strain and can be defined as

$$\boldsymbol{\varepsilon}_{eff}^p = \int_0^t \sqrt{\frac{2}{3}} d\boldsymbol{\varepsilon}_{ij}^p d\boldsymbol{\varepsilon}_{ij}^p. \text{ Once the stress reaches yield strength, the material will}$$

exhibit damage behavior with a damage variable $d_m = 0.9$.

In order to obtain equivalent thermodynamic parameters and damage evolution of carbon yarn more accurately, a bilinear cohesive model governed with a traction-separation law is used to characterize the damage behavior of

fiber/matrix interface. The bilinear models of cohesive element under pure and mixed modes are depicted in Fig. 5a. The relation between cohesive traction (t_n , t_s) and corresponding separations (δ_n , δ_s) in the interface can be expressed as

$$t_i = K_i \delta_i (1 - D_i) \quad (i = n, s) \quad (4)$$

where K_i is cohesive stiffness, which can be defined as $K_i = \frac{t_i^{\max}}{\delta_i^0}$ during initial stage. D_i is damage parameter associated with each mode dominated bilinear cohesive law, which is defined by

$$D_i = \begin{cases} 0 & \delta_i < \delta_i^0 \\ \left(\frac{\delta_i - \delta_i^0}{\delta_i} \right) \left(\frac{\delta_i^f}{\delta_i^f - \delta_i^0} \right) & \delta_i^0 < \delta_i < \delta_i^f \\ 1 & \delta_i > \delta_i^f \end{cases} \quad (5)$$

where δ_i^0 and δ_i^f are separation displacements at maximum cohesive traction and the completion of debonding, respectively. After damage of the interface is activated, the interfacial stiffness will gradually reduce to zero linearly in accordance with Eq. (5). This damage evolution behavior is controlled by fracture energy based on mixed-mode power law criterion with the power law exponent $\alpha = 1.2$, which is defined by

$$\left\{ \frac{G_I}{G_{IC}} \right\}^\alpha + \left\{ \frac{G_{II}}{G_{IIC}} \right\}^\alpha = 1 \quad (6)$$

where G_i and G_{ic} ($i = I, II$) denote the corresponding fracture energies and fracture toughness in normal direction and shear directions, respectively.

Since the lack of interface properties for carbon fiber/matrix (T700/F-46), a set of experimentally calibrated interface parameters of similar material system (T700/TF1408) are used here to describe the change of interface properties with temperature [3]. The kink force F_d and maximum pull-out force F_{\max} measured by microbond test at different temperatures are listed in Table 2.

Fracture toughness G_{IC} varying with temperature is obtained by the relation between F_d and glass transition temperature

$$G_{IC} = \frac{r_f C_{33s}}{2} \left[\frac{F_d}{\pi r_f^2} + \frac{(\alpha_f - \alpha_m) \Delta T_n}{2 C_{33s}} \right]^2 \quad (7)$$

where α_f and α_m are coefficients of thermal expansion of axial fiber and matrix, respectively. ΔT_n is the difference between the ambient and glass transition temperature and C_{33s} is given by

$$C_{33s} = \frac{1}{2} \left[\frac{1}{E_{f1}} + \frac{V_f}{V_m E_m} \right] \quad (8)$$

where V_f and V_m are volume fractions of the T700 carbon fiber and F-46 matrix, respectively. Interface shear strength τ_{IFSS} can be calculated by

$$\tau_{IFSS} = \frac{F_{\max}}{\pi d_f l_e} \quad (9)$$

where is d_f the diameter of fiber and l_e is the embedded length of microdroplet with a value of 70~90 μm . The calculated inter-phase properties at different temperature are listed in Table 3. Due to the absence of detailed experimental data, the critical normal fracture energy G_{IC} and shear fracture energy (G_{IIC} and G_{IIIC}) of the interface are assumed to be equal. Similarly, interfacial normal strength t_n^{\max} and shear strength t_s^{\max} are assumed to be equal.

Based on the proposed microscale classical hexagon model with interface damage of fiber/matrix, the stress-strain curves and failure modes of microscale RVE at 25 °C, 60 °C, 80 °C and 120 °C are shown in Figs. 5c-h. In longitudinal direction, as temperature rises, interface properties of strength and fracture toughness decreases so that the degree of interface slip increases, which is shown in Fig. 5b. However, the longitudinal tensile, compressive stiffness and strength

are slightly changed, because the failure is mainly controlled by temperature-independent carbon fibers. For the transverse tension and compression, stiffness and strength are mainly determined by resin matrix and interface, whose properties vary with temperature. As shown in Figs. 5e-f, stress-strain curves exhibit a trend of first rising and then falling, which corresponds to the interface damage evolution represented by the traction-separation curve of interface. It should be noted that interface failure leads to the final failure of RVE in transverse compression as the temperature rises. For in-plane shear, stress-strain curves present typical elastic-plastic properties and the failure modes become the damage of resin matrix and interface. For out-plane shear, the final failure is governed by the shear failure of resin matrix. The out-plane shear properties decline with the resin softening at higher temperature, and the failure modes at different temperatures are similar.

We undertake major revisions and conduct new analyses to enhance the quality of our study that we outline below: We clarify the potential application of our work in the relevant sections. We distinguish the contributions of thermal effect and electromagnetic effect to the composites, and further reveal the weakening mechanism of electromagnetic impact damage on 3D orthogonal woven composites. Moreover, we explain why the impact tolerance of the composite used in our work decreases with increasing temperature, and consider temperature-dependent fiber/matrix interface properties. We believe that we address all of the reviewer's feedback and that our revisions substantially improve our manuscript.

- [1] He, C.W. A hierarchical multiscale model for the elastic-plastic damage behavior of 3D braided composites at high temperature. *Compos. Sci. Technol* 196, 108230 (2020).
- [2] Sauder, C. et al. Thermomechanical properties of carbon fibres at high temperatures (up to 2000 C). *Compos. Sci. Technol.* 62, 499-504 (2002).
- [3] Wang, H.X. et al. Experimental and numerical study of the interfacial shear strength in carbon fiber/epoxy resin composite under thermal loads. *Int J Polym Sci* 2018, 1-8 (2018).

Reviewer #2: In this work, a pulsed current was applied to woven composite structures to suppress damage from impact.

1. Please proof read manuscript for typographical errors and proper grammar

Answer: Thank you for this suggestion. We tried our best to improve the manuscript and made some changes to the manuscript. And we hope the revised manuscript could be acceptable for you.

2. What is a delta bond and what is the variable pi in the introduction. These terms must be better defined

Answer: Thank you for this suggestion. A detailed explanation of these terms has been added in the revised manuscript.

3. In this work, impact testing with a pulsed current experimental setup was performed. In addition a multi-physics FEA simulation was conducted to compare results. This method was able to reduce inelastic energy and deformation to test samples. In the experimental program, it is not clear how many samples were tested. Sample size, error bars and other statistical data is required to support experimental findings from the impact tests.

Answer: Thank you for this suggestion. Our work aims to rationally control the external electromagnetic field to increase the impact impedance of 3D orthogonal woven composites and to reveal the improvement mechanism. After extensive early testing, we find that pulse current can avoid thermal effects more effectively than direct current. On this basis, we carefully select four samples to conduct a comparative study of the effect of different pulse current peaks on damage. Our findings are consistent with those of Snyder et al. [1] and Barakati et al. [2] for unidirectional laminates. A certain degree electromagnetic environment can decrease delamination and deformation of the exposed carbon fiber/epoxy composites. The dimensions of the specimen have been described in our initial manuscript submission and annotated in Fig. 2b.

4. *The approach outline in the manuscript is interesting but it isn't clear how this method would apply in a real-world setting. In the introduction the authors state that damage to carbon fiber composites is commonly caused by events like tool strikes or bird impacts. From my understanding of the method the timing of the electrical pulse has to coincide with the impact damage event. I'm curious how this method will apply when in the real world timing of damage will be unpredictable and therefore may not occur at the same time of a given electrical pulse. Therefore, I am not sure of the practicality of this method.*

Answer: We appreciate you highlighting the novelty of our study and we appreciate the feedback that the practical application lack clarification. The existing experimental evidence suggested that a certain degree electromagnetic environment can improve the strength and resistance to debris-induced fracture or delamination of the exposed carbon fiber/epoxy laminate. In addition, electromagnetic launch (EML) technology has been used in electromagnetic catapults, electromagnetic railguns, and electromagnetic coilguns for military confrontations. Electric vehicle charging on electrified roads has made use of wireless power transfer technologies. These technologies mainly take advantage of the positive influence of "electromagnetic effect" on conductive solids and the precise control and diagnosis of structural and electromagnetic systems by information flow. As a result, the advanced control techniques can couple the structural capabilities of conductive composites with electrical, magnetic or thermal functions, providing rich possibilities for multifunctional platforms. Inspired by the above research, the purpose of our study is to improve the performance of composite materials to make full use of the multifunctional property of materials, such as a key physical effect in conductive fibers whereby matter becomes compressed under the effect of an electric field. Our work has some potential applications in reducing impact damage from foreign bodies (such as debris, stones, dropped hammer) by pre-monitoring harmful signals and

triggering current excitation.

- [1] Snyder, D. R. Preliminary assessment of electro-thermo-magnetically loaded composite panel impact resistance/crack propagation with high speed digital laser photography. *Proceedings of SPIE-The International Society for Optical Engineering*. 4183, 488-513 (2001).
- [2] Barakati, A., Zhupanska, O.I. Mechanical response of electrically conductive laminated composite plates in the presence of an electromagnetic field. *Compos. Struct.* 113, 298-307 (2014).

Consulting Editor's comments:

1. While the reviewers find your work of interest, they raise substantive concerns that cast doubt on the practical benefit of your work and the strength of the novel conclusions that can be drawn at this stage. In this case, we are concerned about being able to use this system in a real-world setting, for example, with random damage.

Answer: Thank you for this comment. The aim of our work is to improve the impact impedance of 3D orthogonal woven composites by controlling the external electromagnetic field reasonably and to reveal the improvement mechanism. Research by Snyder et al. [1] and Barakati et al. [2] suggested that a certain degree electromagnetic environment can decrease delamination and deformation of the exposed carbon fiber/epoxy laminates. In addition, the technique of precise control and diagnosis of structure and electromagnetic system using information flow has been maturely applied to electromagnetic (EM) catapult, EM railgun, EM coilguns and electric vehicle charging on electrified roads. As a result, the advanced control techniques can couple the structural capabilities of conductive composites with electrical, magnetic or thermal functions, providing rich possibilities for multifunctional platforms. Our work has some potential applications in reducing impact damage from foreign bodies (such as debris, stones, dropped hammer, etc.) by pre-monitoring harmful

signals and triggering current excitation. These explanations are also added in the revised manuscript.

2. *For this type of work, we would need to see more detailed microscopy to analyze the cracking behavior and delamination.*

Answer: Thank you for this suggestion. We fully agree with you about the importance of more detailed microscopy to analyze the cracking behavior and delamination. The gross damage images with a resolution of 1500×1500 pixels are shown in Figs. 3f-i for the plates under impact energy of 50 J with $I_{am} = 0$ A, 30 A, 70 A and 110 A. The results show that the damage area decreases with an increase of pulse current peak. In order to clarify the damage modes of 3DOWCs subject to impact load and guide the selection of material constitutive models in the simulation, we performed microscopic analysis on the plate subject to an impact energy of 50 J (see Fig. 3e). Since damage modes of the 3DOWCs are fixed at the same impact energy, the electromagnetic field acts only as an external field and does not affect the material constitutive models. Based on damage modes observed experimentally, we further use numerical simulation methods for the microscale yarn RVE model to provide more comprehensive detail. See response to reviewer 1 above for additional details.

- [1] Snyder, D.R. Preliminary assessment of electro-thermo-magnetically loaded composite panel impact resistance/crack propagation with high speed digital laser photography. *Proceedings of SPIE-The International Society for Optical Engineering*. 4183, 488-513 (2001).
- [2] Barakati, A., Zhupanska, O.I. Mechanical response of electrically conductive laminated composite plates in the presence of an electromagnetic field. *Compos. Struct.* 113, 298-307 (2014).

May 20 2023

Prof. Fusheng Wang

Assoc. Prof. Chenguang Huang

Dr. Yan Li

School of Mechanics, Civil Engineering and Architecture

Northwestern Polytechnical University

Xi'an, 710129

P.R. China

REVIEWERS' COMMENTS

Reviewer #1 (Remarks to the Author):

I appreciate authors' effort authors for verifying the potential effect of temperature on interface strength and/or fiber/polymer interactions.

The inclusion of interface properties in the multiphysics model and the discussion in the responses clarify my concerns.

Reviewer #2 (Remarks to the Author):

Dear Authors, thank you for addressing the comments raised on the last manuscript draft. I believe these comments have been satisfactorily addressed.

Response to the reviewer and editor comments

We have received your comments of our manuscript (NCOMMS-22-27366A-Z).

We thank the reviewers and editor for their approvals of our revised manuscript.

Reviewers' comments:

Reviewer #1: I appreciate authors' effort authors for verifying the potential effect of temperature on interface strength and/or fiber/polymer interactions.

The inclusion of interface properties in the multiphysics model and the discussion in the responses clarify my concerns.

Answer: Thank you for your approval of our paper.

Reviewer #2: Dear Authors, thank you for addressing the comments raised on the last manuscript draft. I believe these comments have been satisfactorily addressed.

Answer: Thank you for your approval of our paper.

July 26 2023

Prof. Fusheng Wang

Assoc. Prof. Chenguang Huang

Dr. Yan Li

School of Mechanics, Civil Engineering and Architecture

Northwestern Polytechnical University

Xi'an, 710129

P.R. China